Knockdown of hsa_circ_0008922 inhibits the progression of glioma

Xue Chunhong 1
Liu Chang 2 3
Yun Xiang 4
Zou Xiaoqiong 1
Li Xin 1
Wang Ping 1
Li Feng 1
Ge Yingying 1 5
Zhang Qingmei 1 5
Xie Xiaoxun 1 5 6
Li Xisheng 7 lxssky@hotmail.com
Luo Bin 1 5 glbinbin@sr.gxmu.edu.cn
1 Department of Histology and Embryology, School of Basic Medicine Science, Guangxi Medical University , Nanning , China
2 Department of Neurosurgery, The First Affiliated Hospital of Guangxi Medical University , Nanning , China
3 Postdoctoral Research Station, School of Basic Medicine Science, Guangxi Medical University , Nanning , China
4 Department of International Cooperation and External Exchange, The First Affiliated Hospital of Guangxi Medical University , Nanning , China
5 Key Laboratory of Preclinical Medicine (Guangxi Medical University), Education Department of Guangxi Zhuang Autonomous Region , Nanning , China
6 Key Laboratory of Early Prevention and Treatment of Regional High Frequency Tumor (Guangxi Medical University), Ministry of Education , Nanning , China
7 Department of Neurosurgery, The People’s Hospital of Guangxi Zhuang Autonomous Region, Guangxi Academy of Medical Sciences , Nanning , China
Tyagi Abhishek
Electronic publication date: 2022 Dec 20
Publication date: 2022
Volume: 10
Electronic Location ID: e14552
Received 2022 Sep 26; Accepted 2022 Nov 20
Copyright: © 2022 Xue et al.
Copyright year: 2022
Copyright holder: Xue et al.
License: This is an open access article distributed under the terms of the Creative Commons Attribution License, which permits unrestricted use, distribution, reproduction and adaptation in any medium and for any purpose provided that it is properly attributed. For attribution, the original author(s), title, publication source (PeerJ) and either DOI or URL of the article must be cited.
License URL: https://creativecommons.org/licenses/by/4.0/

Keywords: Glioma, circRNA, hsa_circ_0008922, Biological functions, Bioinformatics.

Funding: National Natural Science Foundation of China 81860445, 81960453, 82260554 and 81660429 Natural Science Foundation of Guangxi Province 2022GXNSFAA035639, 2018GXNSFAA050151, 2018GXNSFAA281050, 2018GXNSFAA281251, 2018GXNSFAA050058 and 2018GXNSFBA281187 Sanitation Research Project of Guangxi Z20190132 Science and Technology Plan Project of Qingxiu 2019037 This work was supported by the National Natural Science Foundation of China (No. 81860445, No. 81960453, No. 82260554 and No. 81660429), the Natural Science Foundation of Guangxi Province (No. 2022GXNSFAA035639, No. 2018GXNSFAA050151, No. 2018GXNSFAA281050, No. 2018GXNSFAA281251, No. 2018GXNSFAA050058 and No. 2018GXNSFBA281187), the Sanitation Research Project of Guangxi (No. Z20190132), and the Science and Technology Plan Project of Qingxiu (No. 2019037). The funders had no role in study design, data collection and analysis, decision to publish, or preparation of the manuscript.

==============================
Background

A glioma is a tumor originating from glial cells in the central nervous system. Although significant progress has been made in diagnosis and treatment, most high-grade glioma patients are prone to recurrence. Therefore, molecular targeted therapy may become a new direction for adjuvant therapy in glioma. In recent years, many studies have revealed that circular RNA (circRNA) may play an important role in the occurrence and development of many tumors including gliomas. Our previous study found that the expression of hsa_circ_0008922 was up-regulated in glioma tissues upon RNA sequencing. The biological mechanism of circ_0008922 is still unreported in gliomas. Therefore, in this study, we preliminarily outlined the expression of hsa_circ_0008922 in glioma and explored its biological functions.

Methods

The expression of hsa_circ_0008922 in forty glioma tissues and four glioma cell lines (A172, U251, SF763 and U87) was detected by quantitative real-time polymerase chain reaction (qRT-PCR). The correlation between hsa_circ_0008922 expression and clinicopathological features of glioma patients was evaluated by Fisher’s exact test. To understand the potential function of hsa_circ_0008922 in glioma, we constructed small interfering RNA (siRNA) to hsa_circ_0008922 to downregulate its expression in glioma cell lines A172 and U251. With these hsa_circ_0008922 downregulated cells, a series of assays were carried out as follows. Cell proliferation was detected by CCK8 assay, migration and invasion were determined by wound healing assay and transwell assay, respectively. Colony formation ability was evaluated by plate clonogenic assay. Moreover, flow cytometry combined with Western blot was performed to analyze apoptosis status and the expression of apoptotic related proteins (caspase 3 and caspase 9). Finally, the possible biological pathways and potential miRNA targets of hsa_circ_0008922 were predicted by bioinformatics.

Results

We found that the expression of hsa_circ_0008922 in glioma tissues was 3.4 times higher than that in normal tissues. The expression of has_circ_0008922 was correlated with WHO tumor grade. After down-regulating the expression of hsa_circ_0008922, malignant biological behavior of glioma cells was inhibited, such as cell proliferation, colony formation, migration, and invasion. At the same time, it also induced apoptosis of glioma cells. Predicted analysis by bioinformatics demonstrated that hsa_circ_0008922 may be involved in tumor-related pathways by acting as a molecular sponge for multiple miRNAs (hsa-let-7e-5p, hsa-miR-506-5p, hsa-let-7b-5p, hsa-let-7c-5p and hsa-let-7a-5p). Finally, we integrated our observation to build a circRNA-miRNA-mRNA predictive network.

Introduction

Glioma is one of the most common primary malignant tumors in the central nervous system. Glioblastoma multiforme (GBM) is the highest grade (IV grade) based on the WHO classification and has high invasiveness and lethality (Ng et al., 2020; Sminia et al., 2021). Despite improvements in current treatment modalities (surgery, radiotherapy and chemotherapy) (Nabors et al., 2020), the prognosis of GBM remains poor and there is still a 90% recurrence rate after treatment (Francis et al., 2022; Easaw et al., 2013). This poses a serious threat to the health of patients and brings heavy economic burden to patients and their families. Therefore, it is critical to identify potential molecules for adjuvant therapy option in glioma.

circRNA is a new class of non-coding RNAs with covalently closed-loop structures (Cocquerelle et al., 1993), which is characterized by high stability and high conservation (Zhang, Yang & Xiao, 2018), and is mainly distributed in the cytoplasm. In recent years, a variety of high-throughput sequencing confirmed that many circRNAs were highly expressed in various tissues including tumors (Xu et al., 2019, 2018; Starke et al., 2015). In addition, circRNA can also act as a sponge for microRNAs (miRNA), endogenous competitive RNA to participate in many pathophysiological processes (Sun et al., 2020). Therefore, the expression profile and role of circRNA in glioma can not only provide a new treatment strategy, but also promote the development of precision medicine in clinical diagnosis and treatment.

With RNA sequencing of a panel of glioma tissues and normal brain tissues we found that hsa_circ_0008922 was highly expressed in glioma tissues compared with normal brain tissues. As far as we know, there is only a report for hsa_circ_0008922 showing its abundant expression in hypopharyngeal squamous cell carcinoma and correlation with worse outcome of patients (Wang et al., 2020). However, no studies have verified the function of hsa_circ_0008922 in glioma. In this study, we found that hsa_circ_0008922 expression was significantly associated with tumor grade, tumor size and Ki-67 which is a nuclear protein expressed in all proliferating cells (Røge et al., 2021). The down-regulation of hsa_circ_0008922 expression in vitro can weaken the malignant behavior of glioma cells, such as decreasing cell proliferation, migration, invasion and inducing apoptosis. Additionally, bioinformatic analysis demonstrated that hsa_circ_0008922 may exert a molecular sponge role for some miRNAs in glioma. Therefore, these results provide a basis for further study of hsa_circ_0008922 in glioma in future.

Materials and Methods

Patients and samples

From May 2018 to November 2021, glioma tissues were obtained from 40 patients with glioma who underwent surgical resection in the Department of Neurosurgery, the First Affiliated Hospital of Guangxi Medical University. All patients were confirmed to be glioma by postoperative pathology without preoperative radiotherapy or chemotherapy. At the same time, 10 non-glioma surgical brain tissue samples were collected. The brain tissue was obtained from the brain tissue needed to be removed due to the surgical approach and the focal edema tissue (normal brain tissue), and the histological characteristics were confirmed by the pathologist. All surgically resected specimens were frozen in liquid nitrogen. The specimen collection was examined and authorized by Medical Ethics Committee of Guangxi Medical University (2018087). We received written informed consent from participants of our study.

RNA sequencing

The tissues were collected as previously described in Liu et al. (2021). Specifically, to reduce the impact of individual differences, we set up each group containing five separate samples from five patients. The 10 tissues were divided into two groups: NB (normal brain tissue, n = 5) and GBM (glioblastoma, n = 5). Total RNA was extracted from glioma tissues using TRIzol reagent (Invitrogen, Carlsbad, CA, USA) according to the manufacturer’s protocol and sequenced by Aksomics Biology Technology Co. Ltd (Shanghai, China). The RNA library was constructed by using KAPA Stranded RNA-Seq Library Prep Kit (Illumina, San Diego, CA, USA). The raw sequencing data (FASTQ files generated by the Illumina sequencer) is subjected to quality control to assess whether the sequencing data can be used for subsequent analysis. The expression of circRNA was quantified by calculating the Backsplice junction reads through CIRCexplorer2. Differential expression analysis was conducted using Ballgown (https://rdrr.io/bioc/ballgown/). The circRNA with a P-value ≤ 0.05 and fold change ≥ or ≤1.5 was considered significantly differentially expressed. The statistical power of this experimental design, calculated in RNASeqPower using the web page at https://rodrigo-arcoverde.shinyapps.io/rnaseq_power_calc/ is 0.8083.

Cell culture and transfection

The human GM cell lines (A172, U251, SF763 and U87) were purchased from the Chinese Academy of Sciences (Shanghai, China). Cells were cultured in complete medium (10% fetal bovine serum (Wisent, Quebec, Canada); 1% penicillin streptomycin (Solarbio, Beijing, China); DMEM medium (Wisent, Quebec, Canada)) and placed in a 37 °C, 5% CO2 incubator.

Based on the hsa_circ_0008922 sequence, two siRNAs (s1-hsa_circ_0008922 and s2-hsa_circ_0008922) and random siRNA (NC) fragments were ordered from Bioengineering Co., Ltd. (Shanghai, China). According to the instructions for siRNA usage, Lipofectamine 3000 (Invitrogen, Carlsbad, CA, USA) was used to transfect siRNA into cells. The sequences of siRNA hsa_circ_0008922 were as follows: s 1_hsa_circ_0008922: (sense) 5′-AAG AUA AGU AAC GAU GAC U-3′, (antisense) 5′-AGU CAU CGU UAC UUA UCU U-3′; s2_hsa_circ_0008922: (sense) 5′-UAA CGA UGA CUU GAA AGU A-3′, (antisense) 5′-UAC UUU CAA GUC AUC GUU A-3′.

qRT-PCR

Total RNA was extracted by Vazyme Kit (RC101-01) (Vazyme, Nanjing, China). RNA was reverse transcribed into cDNA by HiScript III RT SuperMix 100 for qPCR (+gDNAwiper) (Vazyme, Nanjing, China). ChamQSYBR qPCR Master Mix (Vazyme Q711-02; Vazyme, Nanjing, China) was used for polymerase chain reaction in StepOne Real-time PCR System (Applied Biosystems, Waltham, MA, USA). The relative level was calculated with 2−ΔΔCt.

RNase R exonuclease digestion experiment and electrophoresis

According to the instructions in RNase R endonuclease digestion kit (Geneseed Biotech Co., Ltd., Guangzhou, China), the total RNA was divided into digestion group (RNase R treatment) and control group, with 5 μg RNA (1 μg RNA required 8 U RNase R endonuclease digestion) in each group. the RNase R was not added in the control group. The above groups were incubated at 37 °C for 25 min. Then 1 μg RNA was reversed and detected by qRT-PCR. The qRT-PCR products amplified by the divergent primer were transcribed to Sangon Biotech (Shanghai, China) Co., Ltd. for TA cloning sequencing to determine the full length of PCR products. The PCR products of gDNA and cDNA were further detected by 1.3% agarose gel electrophoresis. PCR products were separated by 110 V electrophoresis for 35 min and detected by UV. GL DNA Marker 100 (AGbio, Beijing, China) was used as a marker of DNA size. The sequences of primers were as follows: hsa_circ_0008922 Divergent Primer: (sense) 5′-TCC ATC AGG ACC CCA GAT GTC-3′, (antisense) 5′-ACT GCA CAT GCA GAC TGT CAC-3′; hsa_circ_0008922 convergent Primer: (sense) 5′-GGG CAT CCT TCA CCC ATC TG-3′, (antisense) 5′-ATC TTG GTG TCA CAC AGG GC-3′.

Wound healing assay

After transfection with siRNA to hsa_circ_0008922 for 48 h, 100 μL of 2 × 104 cells were lifted and seeded into the single hole of ibidi plug-in and cultured for 22 h. After the cells were full of ibidi plug-in, the plug-in was removed. Then the DMEM complete medium was replaced with appropriate low serum DMEM complete medium (2% FBS). Photographs were taken at 0, 6, 12 h and 0, 12, 24 h, respectively.

CCK8 assay

CCK8 detection kit (Vazyme A311-01; Vazyme, Nanjing, China) was used to detect the proliferation of glioma cells treated with siRNA. A total of 100 μL of 4 × 103 cells were inoculated in 96-well plates, five wells in each group. A total of 10 μL of CCK8 solution was added to each well and cells were incubated at 5% CO2 37 °C for 2 h. The optical density (OD) values at 450 nm were measured at 0, 24, 48, 72 and 96 h, respectively.

Transwell assay

The migration and invasion ability of glioma cells were evaluated using a Transwell chamber (Corning Incorporated, Corning, NY, USA) with or without matrix glue (Absin). Forty-eight hours after transfection, a 200 μL cell suspension (2 × 104 cells) was inoculated in the upper chamber of transwell chamber, and DMEM (500 μL) containing 20% FBS was added in the lower chamber of transwell chamber. They were incubated at 37 °C for 24 h. Then, 4% methanol was applied to fix the cells for 30 min followed by staining with 0.1% Crystal Violet Stain solution (Solarbio, Beijing, China) for 30 min. Stained cells were photographed by inverted microscope (Olympus, Shinjuku City, Japan) and counted manually.

Flow cytometry

Apoptosis was detected by using FITC Annexin V apoptosis detection Kit (BD Biosciences 556547; BD Biosciences, Franklin Lakes, NJ, USA). After glioma cells were transfected, the cells were completely digested with EDTA-free trypsin and washed twice with PBS buffering at 4 °C. The cells were dyed with 5 μL FITC Annexin V and 5 μL propidium iodide (PI) for 15 min at room temperature in the dark and then observed by flow cytometry (BD, Franklin Lakes, NJ, USA). FlowjoV 1.8.1 (Becton, Dickinson & Company, Franklin Lakes, NJ, USA) was used to analyze the results.

Western blot

Western Blot was used to detect apoptosis-related proteins in glioma cells. After transfecting siRNA into cells and culturing cells for 48 h, the cells were harvested and lysed with protease inhibitor and phosphatase inhibitor to extract proteins. Then, the protein was separated by SDS-PAGE and transferred to PVDF membrane. 1:1,000 diluted primary antibodies (caspase 3, rabbit, #14220S; caspase 9, mouse, #9508, Cell Signaling Technology and GAPDH, rabbit, #ab181602 Abcam, Cambridge, UK) were incubated overnight at 4 °C. After completing incubation of primary antibodies, the secondary antibodies in 1:5,000 dilution were incubated at room temperature (rabbit, #ab6721; mouse, #ab6728 Abcam Cambridge, UK).

Plate cloning assay

The clone formation rate indicated the proliferation capacity and cell population dependency (Liu et al., 2022).

3 × 103 of siRNA treated cells were inoculated in the 6-well plates at 37 °C in 5 % CO2 for 11 days. After culture, cells were fixed with 4% formaldehyde for 30 min and stained with 0.1% Crystal Violet Stain solution for 12 min. The photograph was recorded with a stereomicroscope (Olympus, Shinjuku City, Japan).

Bioinformatics analysis

ENCORI (https://starbase.sysu.edu.cn/agoClipRNA.php?source=circRNA) was used to predict hsa_circ_0008922 downstream miRNA. OECloud tools (https://cloud.oebiotech.cn/task/detail/array_miranda_plot/?version=old) was used to predict binding fraction and free energy. miRDB (http://mirdb.org/index.html), miRWalk (http://mirwalk.umm.uni-heidelberg.de/), Linkedomics (http://linkedomics.org/login.php) were used to predict miRNA downstream mRNA, respectively. Cytoscape was used to draw the interaction network of hsa_circ_0008922 with its downstream miRNA, and the interaction network of miRNA with its downstream mRNA (Shannon et al., 2003; Assenov et al., 2008). GO (Gene Ontology) and KEGG (Kyoto Encyclopedia of Genes and Genomes) enrichment analysis of miRNA downstream mRNA was performed on OECloud tools (https://cloud.oebiotech.cn/task/detail/enrichment-oehw/?id=57).

Statistical analysis

Data analysis was performed using Graphpad and SPSS 22.0 statistical analysis software. Student’s t-test was used for comparison between the experimental groups. Fisher’s exact test was used for categorical variables. Quantitative data were expressed as mean ± standard deviation. P < 0.05 was considered statistically significant.

Results

Identifying hsa_circ_0008922 in glioma cell

In order to seek circRNA aberrantly expressed in glioma, we performed RNA sequencing of glioma tissues. Here, we selected a novel circRNA, hsa_circ_0008922, from circRNA identified above because there was no report in glioma except only one study in hypopharyngeal squamous cell carcinoma (Wang et al., 2020). As shown in Fig. 1A, hsa-circ-0008922 is produced from regions of exons 3–4 of (MATR3, NM199189),with a junction of 161 nt (http://www.circbank.cn/). We designed divergent primers for hsa_circ_0008922 and convergent primers for MATR3 mRNA. As shown in Fig. 1B, hsa_circ_0008922 can be amplified by divergent primers with cDNA as template, whereas there was no amplification detected at similar sizes using gDNA as template. Contrary, using convergent primers RT-PCR could amplify MATR3 mRNA in the sample without RNase R treatment, while it failed to amplify MATR3 mRNA in the sample with RNase R treatment. Moreover, PCR products amplified by divergent primers were used for TA cloning and sequencing to confirm that hsa_circ_0008922 had the reverse splicing site (Fig. 1C). These results indicated that the divergent primers we designed can specifically amplify hsa_circ_0008922 and can be used for detection of its expression.

Figure 1 Identify hsa_circ_0008922.

(A) Schematic diagram of the ring formation mechanism of hsa_circ_0008922. (B) Agarose gel electrophoresis was used to detect the stability of hsa_circ_0008922. M is DNA marker. RNase R+ represents RNA treated with RNase.R. RNase R- represents RNA treated without RNase.R. “◀▶” Represents divergent primers. “▶◀” Represents convergent primers. (C) NCBI Blast was used to compare the consistency of PCR product sequencing results with the target sequence. Arrows and yellow line segments indicate circularization sites.

Up-regulation of hsa_circ_0008922 in glioma tissues

We cannot gain any information for the expression profile of hsa_circ_0008922 in glioma from five public databases of circRNA (ClRCpedia v2, http://yang-laboratory.com/circpedia/; circRNADb, circatlas.biols.ac.cn; TSCD, http://gb.whu.edu.cn/TSCD/; MiOncoCirc, https://mioncocirc.github.io/; and circRNADisease, http://cgga.org.cn:9091/circRNADisease/). Therefore, we tested our clinical samples as a primary study for hsa_circ_0008922 expression. As shown in Fig. 2A, the expression of hsa_circ_0008922 in glioma tissues (n = 40) and normal tissues (n = 10) was detected by qRT-PCR. We found that the relative expression of hsa_circ_0008922 was significantly higher in gliomas than in normal tissues (P = 0.027). It was found that in low-grade tumor and high-grade tumor the expression of hsa_circ_0008922 was significantly higher than normal tissues, respectively (Fig. 2B). Furthermore, we also detected the relative expression of hsa_circ_0008922 in different types of gliomas. We found that in astrocytoma and glioblastoma and oligodendroglioma the expression of hsa_circ_0008922 was higher than normal tissues, respectively (Fig. 2C).

Figure 2 hsa_circ_0008922 expression in glioma tissues by qRT-PCR.

(A) The expression of hsa_circ_0008922 in glioma tissues (n = 40) were compared as normal brain tissues (n = 10). (B) The expression of hsa_circ_0008922 was investigated between the groups of normal brain tissues (n = 10), low-grade tumor (n = 20) and high-grade tumor (n = 20). (C) The expression of hsa_circ_0008922 was analyzed in different pathological types of glioma (normal brain tissues n = 10, astrocytoma n = 23, glioblastoma n = 15, oligodendroglioma n = 2. Data is expressed as mean ± SEM. P < 0.05.

Correlation between hsa_circ_0008922 expression and clinicopathological parameters of glioma patients

As hsa_circ_0008922 was up-regulated in gliomas, we further investigated the correlation between the expression of hsa_circ_0008922 and clinical data such as age, gender, tumor size, WHO grade and GFAP (glial fibrillary acidic protein). As shown in Table 1, the expression of hsa_circ_0008922 was correlated with the WHO grade, tumor size and Ki-67 of patients. Multiple linear regression showed that tumor size and WHO grade played a significant role in explaining hsa_circ_0008922 (Table 2).

Table 1 Correlation between hsa_circ_0008922 expression and clinicopathological parameters of glioma patients.

Items		Overall information, n	hsa_circ_0008922 expression, n (%)	P	
Low	High	
Total			20	20		
Gender	Male	23	14 (70)	9 (45)	0.110	
Female	17	6 (30)	11 (55)	
Age (years)	<45	20	13 (65)	11 (55)	0.519	
≥45	20	7 (35)	9 (45)	
Tumor size (diameter)	<5 cm	20	18 (90)	2 (10)	<0.001	
>5 cm	20	2 (10)	18 (90)	
WHO grade	I–II	21	14 (70)	7 (35)	0.027	
III–IV	19	6 (30)	13 (65)	
GFAP	−/+	40	20 (100)	20 (100)	###	
++/+++	0	0 (0)	0 (0)	
P53	−/+	28	15 (75)	13 (65)	0.49	
++/+++	12	5 (25)	7 (35)	
Ki-67 (%)	<20%	22	16 (80)	6 (30)	<0.001	
≥20%	18	4 (20)	14 (70)	
MGMT	–	16	6 (30)	10 (50)	0.197	
+	24	14 (70)	10 (50)	
KPS	<70	14	5 (25)	9 (45)	0.185	
≥70	26	15 (75)	11 (55)	
Notes:

GFAP, glial fibrillary acidic protein; P53, tumor protein p53; Ki67, marker of proliferation Ki-67; MGMT, O6-methyl-guanine DNA methyltransferase; KPS, Karnofsky Performance Status. The relative expression of hsa_circ_0008922 was less than 5.899, which was low expression. And conversely, it was high expression.

### Indicated that statistical analysis cannot be carried out due to data distribution.

Table 2 Univariate and multivariable analysis of prognostic parameters in glioma patients.

Items		Overall information, n	Univariate	Multivariable	
HR	95% CI	P	HR	95% CI	P	
Gender	Male	23	−1.430	[−4.263 to 1.402]	0.313				
Female	17	
Age (years)	<45	20	0.059	[−0.025 to 0.142]	0.163				
≥45	20	
Tumor size (diameter)	<5 cm	20	5.113	[2.823–7.402]	<0.001*	3.933	[2.346–5.519]	<0.001*	
>5 cm	20	
WHO grade	I–II	21	6.098	[4.081–8.116]	<0.001*	3.821	[0.853–6.788]	0.013*	
III–IV	19	
P53	−/+	28	−0.517	[−3.610 to 2.576]	0.737				
++/+++	12	
Ki-67 (%)	<20%	22	6.467	[4.561–8.372]	<0.001*	2.078	[−0.915 to 5.072]	0.168	
≥20%	18	
MGMT	–	16	−2.815	[−5.561 to −0.69]	0.047*	0.168	[−1.521 to 1.858]	0.841	
+	24	
KPS	<70	14	−0.058	[−0.128 to 0.012]	0.104				
≥70	26	
Notes:

Univariate analysis was performed using the simple linear regression; Multivariate analysis was performed using the multiple linear regression. HR, Hazard ratio; 95% CI, 95 percent confidence interval for relative risk.

* P < 0.05.

Screening highly efficient siRNA for hsa_circ_0008922

Considering up-regulation of hsa_circ_0008922 correlated with some of clinicopathological parameters of glioma patients, we further conducted loss-of-function experiments to explore the biological function of has_circ_0008922 in vitro. We first detected the relative expression of hsa_circ_0008922 in a panel of glioma cells (A172, U251, U87 and SF763 cells) by qRT-PCR. Figure 3A shown all of cells expressing hsa_circ_0008922 with different levels. A172 expressed the highest hsa_circ_0008922, which was three times of SF763. Whereas U251 expressed the second higher level of hsa_circ_000892 that was 1.6 times of SF763. Therefore, A172 and U251 were selected for hsa_circ_0008922 depletion cell models.

Figure 3 Identify hsa_circ_0008922 expression in glioma cell lines and select siRNA fragments.

(A) Relative expression of has_circ_0008922 in GM cell lines and measured by qRT‐PCR. (B and C) The hsa_circ_0008922 was down-regulated in A172 and U251 cells, and the knock-down efficiency of the two siRNA fragments was detected at 48 and 72 h. s1-hsa_circ_0008922 and s2-hsa_circ_0008922 represent two siRNA fragments. Data is expressed as mean ± SEM. **P < 0.01, ***P < 0.001, ns indicates no statistically significant difference.

Subsequently, we determined the siRNA fragment and optimized its transfection time in vitro. Transfecting cells with two siRNA fragments, s1-hsa_circ_0008922 and s2-hsa_circ_0008922, respectively, it was found that s2-hsa_circ_0008922 had the best effect of -hsa_circ_0008922 down-regulation in the time point of forty-eight hours (Figs. 3B and 3C). Therefore, we chose s2-hsa_circ_0008922 for hsa_circ_0008922 depletion experiments and set the 48 h after siRNA transfection to perform a series experiment for biological behavior in glioma cells.

Down-regulation of hsa-circ-0008922 inhibits migration and invasion in glioma cells

Because migration and invasion were closely related to the malignant degree of tumor (van de Merbel et al., 2018), we used wound healing assays to evaluate lateral migration of glioma cells. Through the analysis of time series data and comparison between groups, it was found that the migration rate of s2-hsa_circ _0008922 treatment group in A172 (Figs. 4A and 4C) and U251 (Figs. 4B and 4D) was lower than that of the control. Further, the transwell assays of migration and invasion showed suppression of longitudinal migration and invasion of A172 (Figs. 5A–5C) and U251 (Figs. 5D–5F) following hsa-circ-0008922 silencing. In summary, the down-regulation of hsa_circ_0008922 in glioma cells can significantly inhibit the migration and invasion ability of glioma cells.

Figure 4 hsa_circ_0008922 regulates wound healing of A172 and U251 cells.

(A and C) wound healing assay in A172. (B and D) wound healing assay in U251. The ordinate is healing rate. The abscissa is the detection time point. NC: cells transfected with scrambled siRNA s2-has_circ_0008922 represents the second siRNA fragments. Data is expressed as mean ± SEM. **P < 0.01, ***P < 0.001. (scale bar = 100 μm).

Figure 5 hsa_circ_0008922 attenuated migration and invasion in A172 and U251 cells.

(A–C) Migration and invasion assay in A172 (D–F) migration and invasion assay in U251. The vertical axis is the number of cells. NC: cells transfected with scrambled siRNA s2-has_circ_0008922 represents the second siRNA fragments. Data is expressed as mean ± SEM. *P < 0.05, **P < 0.01. (scale bar = 50 μm).

Influence of colony formation, proliferation, and apoptosis after hsa_circ_0008922 depletion in glioma cells

After down-regulating hsa_circ_0008922 in A172 and U251, the colony formation ability of cells was detected by plate cloning. It was found that compared to the NC group, the number of cell colonies in the s2-hsa_circ_0008922 group was significantly lower than that in the NC group (Figs. 6A and 6B). CCK8 experiment was used to detect the proliferation in A172 and U251. In this experiment, we detected cell viability at five time points (0, 24, 48, 72, 96 h). It was found that the cell viability in the s2-hsa_circ_0008922 group in A172 (Fig. 6C) and U251 (Fig. 6D) was significantly different from that in the NC group. This result indicated that the down-regulation of hsa_circ_0008922 in glioma cells can significantly inhibit the cell growth in glioma cells.

Figure 6 Down-regulation of hsa_circ_0008922 inhibited the colony formation and cell viability in A172 and U251 cells.

(A and B) Cell colony formation in A172 and U251. (C and D) Cell viability was detected by CCK8 in A172 and U251 The ordinate is OD value. The abscissa is the detection time point (0, 24, 48, 72, 96 h). NC: cells transfected with scrambled siRNA s2-has_circ_0008922 represents the second siRNA fragments. Data is expressed as mean ± SEM. *P < 0.05, **P < 0.01. (scale bar = 5000 m).

Moreover, flow cytometry was used to study whether hsa_circ_0008922 affected apoptosis in glioma cells. We found that in A172 and U251, the apoptosis rate in the s2-hsa_circ_0008922 group was higher than that in the NC group, and there was significant indigenous difference between the two groups (P < 0.05) (Figs. 7A–7C). Furthermore, we also detected apoptosis proteins caspase 3 and caspase 9. It was found that in A172 (Figs. 7D and 7E) and U251 (Figs. 7D and 7F) the expression of caspase 3 and caspase 9 in the s2-hsa_circ_0008922 group was higher than those in the NC group, respectively. Taken together, hsa_circ_0008922 promoted the proliferation of glioma cells by playing an anti-apoptotic role.

Figure 7 Down-regulation of hsa_circ_0008922 inhibited apoptosis in A172 and U251.

(A–C) Cell apoptosis was detected by flow cytometry in A172 and U251. The ordinate is the apoptosis rate. (D–F) Western blotting analysis of caspase3 and caspase9 levels. The ordinate is the relative protein expression. The abscissa is the detected protein. NC: cells transfected with scrambled siRNA s2-has_circ_0008922 represents the second siRNA fragments. Data is expressed as mean ± SEM. **P < 0.01, ***P < 0.001.

Predicting hsa_circ_0008922-miRNA-mRNA networks with potential functions in glioma by bioinformatics

Generally, circRNA regulates gene transcription and expression by serving as a sponge of miRNA. circRNA-miRNA-mRNA regulatory axis has been widely accepted to illustrate the biological functions of circRNA (Chen, 2016). To understand the potential function of hsa_circ_0008922 in regulation of glioma related genes, we first analyzed miRNAs binding to hsa_circ_0008922 by ENCORI, by which a total of 14 miRNAs (hsa-let-7e-5p, hsa-miR-506-5p, hsa-let-7b-5p, hsa-let-7c-5p, hsa-let-7g-5p, hsa-mir-98-5p, hsa-let-7d-5p, hsa-let-7i-5p, hsa-miR-377-3p, hsa-mir-605-3p, hsa-let-7a-5p, hsa-let-7f-5p, hsa-miR-4458 and hsa-miR-4500) were obtained (Fig. 8A). Then, OECloud tools were used to predict the binding fraction and free energy of hsa_circ_0008922 with its downstream miRNAs. The information of these miRNAs was presented in Table 3. The miRNAs (hsa-let-7e-5p, hsa-miR-506-5p, hsa-let-7b-5p, hsa-let-7c-5p and hsa-let-7a-5p) with length greater than 22 bp, predicted score greater than 150 and predicted free energy less than −15 Kcal/Mol were selected for further bioinformatic analysis according to the OECloud tools at https://cloud.oebiotech.com/task/detail/array_miranda_plot/ (energy is less than 10, the energy is smaller, the accuracy is higher. Score is greater than 140, the score is the greater, the accuracy is the higher. In order to screen out more accurate downstream genes, the screening criteria were appropriately improved). With the combined analysis of three databases (miRDB, miRWalk and Linkedomics) we gained mRNAs that were all related to glioma. As shown in Fig. 8B, 103 mRNAs interact with hsa-let-7a-5p, 123 mRNAs interact with hsa-let-7b-5p, 99 mRNAs interact with hsa-let-7e-5p, 37 mRNAs interact with hsa-let-7c-5p, and 5 mRNAs interact with hsa-miR-506-5p. Finally, an interactive network between miRNA and mRNA was constructed (Fig. 8C).

Figure 8 Prediction of the networks of circRNA-miRNA-mRNA.

(A) The interaction network between hsa_circ_0008922 and its downstream miRNAs was constructed. (B) Venn plots of mRNAs interacting with miRNAs downstream of hsa_circ_0008922 were drawn. (C) The interaction network showed the target mRNAs of five highest-ranking miRNAs matched has_circ_0008922.

Table 3 Prediction of binding sites between miRNA and hsa_circ_0008922.

miRNA	Target	Score	Energy	miRNA length	Target positions	
hsa-let-7e-5p	hsa_circ_0008922	158	−16.84	22	91	
hsa-miR-506-5p	hsa_circ_0008922	155	−16.57	22	76	
hsa-let-7b-5p	hsa_circ_0008922	154	−17.03	22	91	
hsa-let-7c-5p	hsa_circ_0008922	154	−15.93	22	91	
hsa-let-7g-5p	hsa_circ_0008922	154	−14.88	22	91	
hsa-mir-98-5p	hsa_circ_0008922	150	−12	22	91	
hsa-let-7d-5p	hsa_circ_0008922	150	−14.79	22	91	
hsa-let-7i-5p	hsa_circ_0008922	147	−13.24	22	88	
hsa-miR-377-3p	hsa_circ_0008922	143	−13.37	22	126	
hsa-mir-605-3p	hsa_circ_0008922	157	−14.73	22	71	
hsa-let-7a-5p	hsa_circ_0008922	158	−15.82	22	91	
hsa-let-7f-5p	hsa_circ_0008922	150	−12	22	91	
hsa-miR-4458	hsa_circ_0008922	156	−19.13	19	93	
hsa-miR-4500	hsa_circ_0008922	146	−11.62	17	93	
Note:

Score, Score predicted by miranda; Energy, The free energy predicted by miranda; Target Positions, The binding initiation site on hsa_circ_0008922.

Based on the ceRNA hypothesis (Whereas micrornas are known to cause gene silencing by binding mrnas, cernas can competitively bind micrornas to regulate gene expression (Salmena et al., 2011)), namely, mRNAs should be negatively correlated with targeted miRNAs and positively correlated with circRNA. The top 30 of mRNAs fitting this criterion were selected for further functional analysis. GO analysis showed that the functions of these mRNAs were mainly related to multiple signaling pathways, such as mitotic sister chromatid cohesion, stem cell population maintenance, negative regulation of intrinsic apoptotic signaling pathway in response to DNA damage by p53 class mediator, negative regulation of cAMP-mediated signaling, and regulation of transcription involved in G1/S transition of mitotic cell cycle (Fig. 9A and Table S1). KEGG analysis indicated that these mRNAs were involved in signaling pathways regulating pluripotency of stem cells, cell cycle, and microRNAs in cancer and glioma (Fig. 9B and Table S2).

Figure 9 Analysis of the function of mRNAs negatively correlated with miRNAs.

(A) A bubble plot of the top 30 enriched GO terms in mRNAs negatively correlated with miRNAs downstream of hsa_circ_0008922 was drawn. The y-axis represents the enrichment of the top 30 items, and the x-axis represents the enrichment score. The size of the bubbles indicates the number of mRNAs in the GO term. Bubble colors represent P-values. (B) A bubble plot of the top 30 enriched pathways in mRNAs negatively correlated with miRNAs downstream of hsa_circ_0008922 was drawn. The y-axis represents the enrichment of the top 30 items, and the x-axis represents the enrichment score. The size of the bubbles indicates the number of mRNAs in the pathway. Bubble colors represent P-values.

Discussion

Glioma is a very common and aggressive intracranial tumor, which leads to brain dysfunction in many patients and poses a serious threat to the health of patients. At present, the traditional treatment of glioma is the comprehensive treatment in surgery, radiotherapy, and chemotherapy, but the prognosis is still poor, and the postoperative recurrence rate is as high as 90% (Ostrom et al., 2014; Van Meir et al., 2010; DeAngelis, 2001). However, the pathogenesis and the progression of glioma still remain unclear. Therefore, it is very important to understand the related molecules for their expression features and their mechanisms in the occurrence and development of glioma.

In early studies, circRNA was considered as “noise” and had no biological function. However, in recent years, it was found that circRNAs mostly distributed in the brain and had significant biological functions, especially in cancer (Shahzad et al., 2021; Yang et al., 2018; Rybak-Wolf et al., 2015). In 2019, Xia et al. (2019) showed that AKT3-174aa encoded by circ-AKT3 could inhibit the carcinogenicity of glioma mother cells. In 2020, He et al. (2020) reported that circ-MAPK4 was up-regulated in glioma, but its expression in high-grade glioma was significantly higher than that in low-grade glioma. At the same time, Chen et al. (2020) found that the expression level of hsa_circ_0074026 was significantly increased in the parent cell samples and cells of glioma, and was correlated with clinical parameters such as tumor size, WHO grade and overall survival. All of these reports imply circRNA has potential for diagnostic and therapeutic application in gliomas.

Through RNA sequencing, we found that hsa_circ_0008922 highly expressed in glioma tissues. hsa_circ_0008922 is derived from exon 3 and exon 4 of Matrin3 (MATR3). MATR3 is a host gene of hsa_circ_0008922 and it is derived from human chromosome 5 and highly expresses in the brain (Coelho et al., 2016). MATR3 participates in many cellular processes, including binding and stabilizing mRNA, regulating mRNA nuclear output, chromatin remodeling, RNA processing, transcription, translation, and apoptosis (Wang et al., 2020; Salem et al., 2021). In addition, MATR3 has been widely studied in amyotrophic lateral sclerosis (You et al., 2022; Cavalli et al., 2021; van Bruggen et al., 2021). However, the research progressed on circRNA from MATR3 in glioma remains unreported. In this study, we first verified that the expression of hsa_circ_0008922 in glioma tissues was higher than that in normal tissues. Statistical analysis showed that the expression of hsa_circ_0008922 was significantly correlated with WHO grade, indicating that hsa_circ_0008922 could be a significant role in the progression of malignant glioma. The expression of hsa_circ_0008922 was significantly correlated with tumor size and Ki-67, which indicated that hsa_circ_0008922 could have a significant function in proliferation of glioma cells. Down-regulation of hsa_circ_0008922 expression in glioma cell lines A172 and U251 can significantly inhibit the proliferation, migration, and invasion of the two cells, indicating that hsa_circ_0008922 may become a new molecule for molecular targeted therapy.

Previous studies have shown that circRNA is often used as a miRNA sponge to regulate the expression of miRNA in various cancers, forming a circRNA-miRNA-mRNA network to regulate the expression of downstream target genes and affect the development of cancer (Hansen et al., 2013; Kristensen et al., 2018). For example, hsa_circ_0003222 promoted stem cell differentiation and progression of non-small cell lung cancer through miR-527 (Li et al., 2021). Therefore, in this study, we predicted miRNAs regulated by hsa_circ_0008922 through bioinformatics analysis, and selected five miRNAs for bioinformatics prediction based on predicted binding fraction and free energy. Previous studies have shown that hsa-let-7b-5p, hsa-let-7e-5p and hsa-let-7c-5p were all tumor suppressors. In 2019, Xi et al. (2019) found that hsa-let-7b-5p could inhibit the migration, invasion, and cell cycle of glioma cells. Another study found that hsa-let-7e-5p as a tumor suppressor could inhibit the progression of head and neck squamous cell carcinoma by targeting CCR7 expression (Wang et al., 2019). A study reported by Fu et al in 2017 demonstrated that hsa-let-7c-5p may act as a tumor suppressor of breast cancer by negatively regulating ERCC6, providing a new strategy for breast cancer treatment (Fu et al., 2017). Thus, the interaction between the hsa_circ_0008922 and these three members of the hsa-let-7 miRNA family may broaden our view for their functions in glioma.

It was interesting that a gene called DNA polymerase lambda (POLL) may be regulated by above mentioned three miRNAs (hsa-let-7b-5p, hsa-let-7e-5p and hsa-let-7c-5p), simultaneously, and its expression may be negatively correlated with the expression of these three miRNAs. Now, there is a report that POLL is differentially expressed in IDH (lactate dehydrogenase) mutant and wild type carriers, relating to the survival rate of patients with LGG (Pang et al., 2020). In future, a series of experiments such as RIP (RNA Binding Protein Immunoprecipitation Assay), RNA pull-down assay and luciferase reporter gene assay have to be performed to illustrate the complicate function of hsa_circ_0008922 in this circRNA-miRNA-mRNA network.

We performed GO analysis and KEGG analysis on the target genes negatively correlated with five miRNAs, and the results showed that hsa_circ_0008922 regulates cancer-related functions and signaling pathways by regulating several miRNAs to play a carcinogenic role. GO analysis showed that the enriched functions of mRNA negatively correlated with miRNAs were mainly related to cell cycle, including regulation of transcription involved in G1/S transition of mitotic cell cycle and mitotic sister chromatid cohesion. Enriched functions were demonstrated in relating to apoptosis, including negative regulation of intrinsic apoptotic signaling pathway in response to DNA damage by p53 class mediator. As known to all, p53 pathway includes DNA repair, cell-cycle arrest, senescence and apoptosis (Riley et al., 2008). The 53 tumor suppressor gene is among the best and longest studied genes in glioblastoma. Since the key function of the gene product p53 is to arrest cells in G0/1 or trigger apoptosis in response to genotoxic stress, so restoration of p53 function by various methods has been extensively studied. However, drugs designed to facilitate the refolding of the mutant protein into a wild-type conformation have not been successful (Rhun et al., 2019). In the result of KEGG analysis, enriched pathways of mRNA negatively correlated with miRNAs were mainly related to tumor, e.g., microRNAs in cancer signaling pathway, glioma signaling pathway. It was also included in cell cycle signaling pathway, Ras signaling pathway which is a pathway closely related to glioma, activating RAS mutation to increase gliogenesis (Breunig et al., 2015). Therefore, we assumed that p53 pathway and Ras signaling pathway might be potential signaling pathways, in which hsa_circ_0008922 may play an oncogenic role in growth and progression of glioma. Nevertheless, these hypotheses should be further validated in our future studies.

There are several limitations in our study. First, the clinic data of this study lacked overall and disease-free survival. However, currently most of the specimens were collected in 2021, less than half a year before the start of this study, so we cannot get the accurate survival time. We will continue to follow up this batch of specimens. Additionally, to confirm the roles of hsa_circ_0008922, in vivo experiments should be further conducted in the mouse models those we have established. Therefore, the roles of hsa_circ_0008922 in glioma warrant further investigation in our future studies.

Conclusion

Glioma highly expressed hsa_circ_0008922, which could affect the proliferation, migration, and apoptosis of glioma cells. hsa_circ_0008922 may server as a sponge for multiple miRNAs.

Supplemental Information

Supplemental Information 1 The enriched top 30 GO terms.

Raw data for Figure 9.

Click here for additional data file.

Supplemental Information 2 The enriched top 30 KEGG pathways.

Raw data for Figure 9.

Click here for additional data file.

Supplemental Information 3 Reads comparison result statistics.

Raw Pairs: Number of fragments generated from raw sequencing data. Trimmed: Number of fragments (read pairs) resulting from removing 5′ and 3′ splice sequences and filtering out excessively short fragments ≤20 bp. Mapped: reads are mapped to the reference genome. mtRNAs: Ratio of the number of fragments aligned to the mitochondrial genome to the number of Pairs. rRNAs: Ratio of the number of fragments matched to the rRNA to the number of Pairs. Unmapped: reads are not mapped into the reference genome.

Click here for additional data file.

Supplemental Information 4 All Differentially Expressed circRNA.

Click here for additional data file.

Supplemental Information 5 hsa_circ_0008922 TA cloning and sequencing.

PCR products amplified by divergent primers were used for TA cloning and sequencing to confirm that hsa_circ_0008922 had the reverse splicing site

Click here for additional data file.

Supplemental Information 6 Prediction of binding sites between miRNA and hsa_circ_0008922.

Raw data for Figure 8.

Click here for additional data file.

Supplemental Information 7 hsa_circ_0008922-miRNA-mRNA networks.

Raw data for Figure 8.

Click here for additional data file.

Supplemental Information 8 Cell line experiments and WB.

Click here for additional data file.

Supplemental Information 9 qRT-PCR & cell line experiments, and clinicopathologic characteristics (Figures 1–7 & Tables 1 & 2).

Click here for additional data file.

Additional Information and Declarations

Competing Interests

Author Contributions

Human Ethics

DNA Deposition

Data Availability

The authors declare that they have no competing interests.

Chunhong Xue conceived and designed the experiments, performed the experiments, analyzed the data, prepared figures and/or tables, authored or reviewed drafts of the article, and approved the final draft.

Chang Liu conceived and designed the experiments, prepared figures and/or tables, and approved the final draft.

Xiang Yun conceived and designed the experiments, prepared figures and/or tables, and approved the final draft.

Xiaoqiong Zou performed the experiments, prepared figures and/or tables, and approved the final draft.

Xin Li performed the experiments, prepared figures and/or tables, and approved the final draft.

Ping Wang performed the experiments, authored or reviewed drafts of the article, and approved the final draft.

Feng Li performed the experiments, authored or reviewed drafts of the article, and approved the final draft.

Yingying Ge conceived and designed the experiments, analyzed the data, authored or reviewed drafts of the article, and approved the final draft.

Qingmei Zhang conceived and designed the experiments, analyzed the data, authored or reviewed drafts of the article, and approved the final draft.

Xiaoxun Xie conceived and designed the experiments, analyzed the data, authored or reviewed drafts of the article, and approved the final draft.

Xisheng Li conceived and designed the experiments, analyzed the data, authored or reviewed drafts of the article, and approved the final draft.

Bin Luo conceived and designed the experiments, analyzed the data, prepared figures and/or tables, and approved the final draft.

The following information was supplied relating to ethical approvals (i.e., approving body and any reference numbers):

The specimen collection was examined and authorized by Medical Ethics Committee of Guangxi Medical University (2018087).

The following information was supplied regarding the deposition of DNA sequences:

The group II intron/IEP sequences are available at GenBank: OP541861.

The following information was supplied regarding data availability:

The Figs. 8 and 9 are available in the Supplemental Files 4 and 5.

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
