# Peer review of "Knockdown of hsa_circ_0008922 inhibits the progression of glioma"

_PeerJ, doi:10.7717/peerj.14552_

## Round 0.1 · original submission · Minor Revisions

Dear Dr. Luo

Thank you for submitting your manuscript " Knockdown of hsa_circ_0008922 inhibits the progression of glioma" to PeerJ. We have now sufficiently received reports from reviewers who find the study interesting in terms of circRNA in GBM. Therefore, after careful consideration, we have decided to invite a minor revision of the manuscript.

As you will see from the reports copied below, the reviewers raise important concerns regarding circRNA's role in overall survival and clinical validation using the TCGA database. We find that these concerns limit the strength of the study, and therefore we ask you to address all of the reviewers' comments with additional work. Without substantial revisions, we will be unlikely to send the paper back for review.

Important:
If you feel that you are able to comprehensively address the reviewers’ concerns, please provide a point-by-point response to these comments along with your revision. Please show all changes in the manuscript text file with track changes or color highlighting. If you are unable to address specific reviewer requests or find any points invalid, please explain why in the point-by-point response.

Best regards,

Abhishek Tyagi, PhD
Academic Editor,

·

Basic reporting

The manuscript is well-written and well-organized.

Experimental design

manuscript Title: "Knockdown of hsa_circ_0008922 inhibits the progression of glioma (#76739)"

1."Correlation between hsa_circ_0008922 expression and clinicopathological parameters of glioma patients"
The authors mentioned "clinical data such as age, gender, tumor size, WHO grade and GFAP (glial fibrillary acidic protein)" These features are
more general features, authors need to used some pathological information, gener expression correlation information.

2. Did the authors checked is there any correlation between overall and Disease-free survival? It would be better if the authors provide this information.

3. The authors mentinoed "After down-regulating hsa_circ_0008922 in A172 and U251, the colony formation ability of
cells was detected by plate cloning. It was found that compared with the NC group, the number
of cell colonies in the s2-hsa_circ_0008922 group was significantly lower than that in the NC
group (Fig. 6A and B)". The question is how significant they are? any information about statistical significance?

4."The miRNAs (hsa-let-7e-5p, hsa-miR-506-5p, hsa-let-7b-5p, hsa-let-7c-5p
and hsa-let-7a-5p) with length greater than 22 bp, predicted score greater than 150 and predicted
free energy less than - 15". How the authors interpret the data, especially >150 and <-15 are good. what is the unit of energy kcal? or ? the authors need to describe this in detail.


5. How DNA damage is directly or indirectly correlated with selected hsa_gene(s)? it is not clearly explained in this manuscript.
This manuscript is touched several things but not clearly explained any one selected research interest. The authors need to properly discuss the main finding of this study along with supporting information.

6. "In summary, this is first time to report the up-regulated miRNA/hsa_circ_0008922 in glioma and use
436 bioinformatics to analyze the possible biological processes and signaling pathways regulated by
437 hsa_circ_0008922." This statement is superficial, the authors need to confirm before they claim and use the statement like "the first time to report"

7.It will be better to show some large scale data, by using TCGA or other cancer resources to show the expression of selected genes.

8. Discussion part is weak, authors need to focus their findings and justify/validate with suitable citations.
9.Table and 2 almost carry similar information. The authors may consider to combine both tables into one.

Validity of the findings

Has some novelty. There is a lack of interpretation and the authors touched on several points but most of them are very shallow findings.

How DNA damage is directly or indirectly correlated with selected hsa_gene(s)? it is not clearly explained in this manuscript.
This manuscript is touched on several things but not clearly explained any one selected research interest. The authors need to properly discuss the main finding of this study along with supporting information.

"In summary, this is the first time to report the up-regulated miRNA/hsa_circ_0008922 in glioma and use
436 bioinformatics to analyze the possible biological processes and signaling pathways regulated by
437 hsa_circ_0008922." This statement is superficial, the authors need to confirm before they claim and use the statement like "the first time to report"

It will be better to show some large-scale data, by using TCGA or other cancer resources to show the expression of selected genes.

Additional comments

The discussion part is weak, authors need to focus on their findings and justify/validate them with suitable citations.
Table and 2 almost carry similar information. The authors may consider combining both tables into one.

Reviewer 2 ·

Basic reporting

• The study is well executed but the writing in the entire document would highly benefit from the help of a linguist. Most of the sentences are poorly structured and I have highlighted some of the sections with such concerns in the hard copy.
• It would be important that the language is sufficiently improved prior to publication

Experimental design

• The study design is relatively good but the study does not indicate any benefits accrued from study participation for the human subjects whose samples were included in the study. By mentioning that the subjects did not receive any prior radiotherapy or chemotherapy; it becomes important to clearly indicate the measures taken to ensure that the approach was not a deliberate attempt to deny the patients valuable care for the sake of the study.
• The study replication is not clearly communicated “To achieve five repeated samples at each group, a sample was taken from each of one patient to reduce the impact of individual differences” Does this mean 5 samples were taken from each patient? The replication approach needs to be clearly stated.
• For the wound healing assay; were the cells lifted after transfection for introduction into the Ibidi chambers? This is not clear

Validity of the findings

• The study findings are good, however, it is rather cumbersome to read the separate figure legends and the duplicated legend prior to the figure: Is this a journal requirement or the authors’ own preference?
• It would be appropriate to include a ladder with molecular weights for the Western blot in figure 7.
• In addition some figures depicting images do not have scale dimensions (Figure 4 A controls, 5 D controls, figure 6 A and B. Please provide scale dimensions for all figures.

Additional comments

• In general, the study has good findings that can be adopted if grammatically well communicated.

Annotated reviews are not available for download in order to protect the identity of reviewers who chose to remain anonymous.

·

Basic reporting

In this work, authors identified a new circRNA and connected its upregulation to glioma through a variety of experiments that measure different aspects of glioma progression. This is an interesting study introducing a new intervention opportunity for glioblastoma multiforme. Overall, the work is nicely conducted and the conclusions are well drawn. The last part about GO annotation analysis is not very convincing though.

Overall, the study is well conducted, uses professional English for the most part. A few locations that require corrections are highlighted in the annotated version attached with my comments.

Article structure, figures, tables are all appropriate. Raw data is provided.

Experimental design

Methods are well described, except details about the RNA sequencing. Authors could add a table about the mapping statistics. How many reads obtained after sequencing (FASTQ), how many mapped (single mapped, duplicates), how were duplicate mappings treated etc.

Also, authors did not go into any detail after they identified enriched pathways (figure 9). It could be worthwhile to provide some context about the pathways which are most enriched with regards to GBM. Are these pathways well-known in GBM? Are there any drugs in use against hub genes in these pathways? Authors could either discuss these points in the discussion with references or provide results from their own experimentation where they test the activity of these pathways on the GBM patient samples.

Validity of the findings

Statistics is well described. Authors could discuss a bit more about the observed enriched pathways in the discussion.

Additional comments

Additional Comments:
1. Lines 229-230: Authors mention 244 up and 655 down regulated circRNAs were identified after RNASeq. Is their any reason with regards to the biology of circRNAs in glioma that could explain identification of more downregulated than upregulated circRNAs?
2. Lines 235-236: Why does the length of circRNA not fixed? Why does it vary between 92-252bp? Are there any functional implications of shorter or longer circRNAs?
3. Figure 1: Is it known whether MATR3 is differentially expressed in glioma? If yes, is there a connection between the expression of circRNA under study here and MATR3 expression?
4. In Figure 2A, what grade the tumor sample represents?
5. Table 2: This correlation analyses of circRNA expression and clinicopathological features is interesting. Did authors consider performing this analysis by using combination of clinicopathological features and then checking the correlation with circRNA? For example, high tumor grade in males of age >60 years and other such combinations. If additional significant correlation patterns are found, this can shed interesting light into this connection. Alternatively, authors can also attempt to perform a linear regression analysis where circRNA is dependent variable (continuous as well as categorical – hi, lo expression) and clinical features as independent variables. This can tell which clinical features contribute how much to explain the circRNA expression.
6. Lines 332-333: Please explain the rationale for selecting miRNAs with predicted score > 150 + binding energy < -15. Why were these cutoffs selected? A couple of examples showing control miRNAs that have score > 150, but the binding energy is > 15 and vice versa. Authors can then show if they would have selected these control miRNAs, they would not have come to the same conclusion as described in the article.

---

## Round 0.2 · accepted · Accept

Dear Dr. Xue,

We are delighted to accept your manuscript, entitled "Knockdown of hsa_circ_0008922 inhibits the progression of glioma," for publication in PeerJ.

Thank you for choosing to publish your interesting work with us.


With kind regards,
Abhishek Tyagi
Academic Editor, PeerJ

·

Basic reporting

no comment

Experimental design

no comment

Validity of the findings

no comment

·

Basic reporting

Authors have addressed all my questions sufficiently. I have no further issues remaining.

Experimental design

Authors have addressed all my questions sufficiently. I have no further issues remaining.

Validity of the findings

Authors have addressed all my questions sufficiently. I have no further issues remaining.

Additional comments

Authors have addressed all my questions sufficiently. I have no further issues remaining.